# From fear of infection to awareness against stigma: A mixed-methods analysis of discourses on HIV in a parliamentary context

Paule Gonzalez-Recio[1,2], Raquel Barba-Sánchez[1*], Édel Granda[1],
Juan-Miguel Guerras[2], Sara Moreno-García[3], Alex Iglesias[1],
María José Fuster-RuizdeApodaca[4,5], Esther García Expósito[6], David Palma[2,7],
María José Belza[1,2]

1 National Health School, Institute of Health Carlos III (ISCIII), Madrid, Spain, 2 CIBER Epidemiology and Public Health (CIBERESP), Madrid, Spain, 3 Department of Preventive Medicine, Severo Ochoa University Hospital, Leganés, Spain, 4 Department of Social and Organizational Psychology, Universidad Nacional de Educación a Distancia (UNED), Madrid, Spain, 5 Spanish Interdisciplinary AIDS Society (SEISIDA), Madrid, Spain, 6 Department of Preventive Medicine, Gregorio Marañón University Hospital, Madrid, Spain, 7 Department of Epidemiology, Barcelona Public Health Agency, Barcelona, Spain

* rbarba@isciii.es

## Abstract

### Background and purpose

Parliamentary discourse holds significant relevance both socially, due to its impact on stigmatization, and because of its potential legislative consequences. However, despite the persistent stigma surrounding HIV and the numerous regulations that affect people living with HIV, there is a global lack of research on how this topic is addressed in parliamentary debates. This study provides a case analysis from the Madrid Regional Parliament, offering insights that may be applicable to other parliamentary contexts. The principal objective is to analyze the content of HIV-related initiatives in this context, along with its possible relationship to the political parties' ideology.

### Methods

A mixed-methods quantitative and qualitative study was conducted, including all parliamentary initiatives on HIV from the 12th Legislature of the Madrid Regional Parliament (2021–2023). Subthemes, initiative types, parliamentary groups ideologies, and debate dates were analyzed. Additionally, a Critical Discourse Analysis of the interventions in the plenary sessions was carried out and categorized based on the ideologies of the parliamentary groups, offering insights that may be reflective of broader political discourse trends.

**Data availability statement:** The data that support the findings of this study are available in Madrid Assembly's website at https://www.asambleamadrid.es/actividad/iniciativas. The exact analyzed dataset are the 40,642 parliamentary initiatives of the XII Legislature. In order to access it, in the searcher available at https://www.asambleamadrid.es/actividad/iniciativas "Legistura XII" must be selected in the field "Legislatura".

**Funding:** The author(s) received no specific funding for this work.

**Competing interests:** The authors have declared that no competing interests exist.

## Results

In the analyzed legislature, 0.09% of all the initiatives addressed HIV. Of these, 83.3% were written, and only 16.7% were presented orally in plenary sessions. Two-thirds focused on prevention, while those concerning access to treatment accounted for 16.7%, and those addressing stigma made up 11.1%. All initiatives were presented by left-wing or center-left parliamentary groups. Right-wing groups discourses centered on fear and the perception of risk, with a strongly stigmatizing rhetoric. In contrast, left and center-left groups emphasized discrimination and stigma as issues and advocated for universal treatment access.

## Conclusions

HIV is minimally addressed in the Madrid Regional Parliament, and when it is, the focus is more on prevention than on addressing the needs of people living with HIV. The study highlights the significant role of political ideology in shaping parliamentary discourse, with stigmatizing rhetoric mainly present in right-wing groups. These findings may offer insights for other legislative bodies where HIV-related stigma and political ideologies intersect.

## Introduction

Social representations are, above all, systems of values, ideas, and practices that guide and facilitate the exchange among the different social and material aspects of the world around us [1]. In this sense, Durkheim differentiates between individual representations and collective representations; in a way that societies are maintained through collective consciousness, a normative and common knowledge among their members that is irreducible to individual consciousness [2].

The social representation theory allows for the analysis of the development of behaviors and communication among individuals and groups. It is conceived as an exchange through which experiences and theories are qualitatively modified in both their scope and content over time and in different contexts. Communication is more than a mere transmission of messages; it involves differentiation, translation, and combination of its contents. Groups invent, differentiate, and interpret social objects and the representations of other groups [3], interpreting phenomena like illness. Therefore, from this framework, and by employing appropriate techniques, we can approach the fields of representation, communications, and their mediators.

Political ideology, can then be understood not merely as a preference on certain policies, but as a systemic symbolic tool that shapes social meanings using policy and discourse [4,5]. Political ideologies deeply influence how social issues are framed and how stigma is (re)produced or challenged and can have a profound impact on health.

Political parties are also a fundamental part of modern democracies and the production of political discourses and identity politics. According to partisan theory, the

ideological orientation of political parties shapes their positions, priorities and communicative strategies, while partisan attachments have the ability to affect reasoning, perceptions and policy evaluation by individuals [6,7]. Parties ideology not only determines the priorities of politics, but also how policy is brought to the political arena, determining, for example, if health is framed as a matter of human rights, the result of individual choices and behavior, or as a threat to public safety [8,9].

Left-wing parties are more likely to prioritize key public health issues, such as equity, social justice and social determinants of health, while right-wing parties are more like to understand health from a neoliberalist perspective emphasizing individual responsibility, national security and biomedical aspects of health [10,11]. Growing far-right policies have been pointed out because of how their neoliberalist, authoritarian, anti-science and anti-equality ideology impact public health maintaining privileges of those in power and worsening inequalities [12,13]. Thus, political ideology and discourse not only represent ideas and thoughts, but have real consequences on the lives and health, particularly, of vulnerable populations.

In parliamentary contexts, the main arena of political parties, ideology driven discourses are particularly relevant because of their potential legal and normative repercussions, and their dual role in reflecting and influencing public discourse and constructing collective representations and understandings of health issues [13], such as HIV.

Since the 1980s, HIV has profoundly affected the lives of numerous people and communities, as well as public life, quickly being conceptualized as both a social and public health issue [14]. However, it was towards the end of that decade, and increasingly so from the 1990s onward, that discourses aiming to reduce discrimination and stigma gained strength. Initially, rooted in activist groups such as Act Up [15] or Radical Gai in Spain, these discourses became part of academic works [16].

Since then, health inequalities, stigmatizing representations surrounding HIV, and their interconnection with social, economic, and political processes have been the subject of several studies [17–19]. Additionally, cognitive frameworks generated in public spaces and their correlation with reactions to HIV have been analyzed [20], as well as the potential contribution to stigma by certain public health campaigns [21], and the global symbolic construction of HIV across different time periods [22].

The HIV-related stigma response is currently central to the UNAIDS strategy [23]. Evidence suggests that ending stigma is crucial not only for the care of people living with HIV [24] but also to effectively end the AIDS epidemic [25]. Therefore, it is necessary to approach HIV research not only from a biomedical perspective but also by recognizing its dimension as a social and political phenomenon characterized by different meanings, values, beliefs, and feelings that have material consequences on the lives of affected communities.

As Sontag points out, the political and discriminatory use of metaphors about HIV, as well as its portrayal as a political symbol or allegory, has devastating consequences for those living with HIV [26]. Most of the previous research on institutional discourses around HIV has focused on public health governance and campaigns. Institutional discourses initially focused on the control of vulnerable populations and blame attribution, adding weight to HIV stigma [20]. According to previous research, more recent institutional public health campaigns have used fear and framed HIV as a threat to security [27] in a similar way of how right-wing politicians prioritize security and individual responsibility while ignoring social determinants of health [10,11,13].

However, although —as previously acknowledged— analyses of institutional discourses around HIV have been conducted, they fail to focus on political ideology of government institutions. Despite the significant relevance of parliamentary settings in modern democracies, there remains a lack of studies that specifically analyze HIV related discourses in this context and their relationship with the ideology of different political parties, which is patent in other public health matters [10,11,13]. It is also important to consider that, in the current rise of far-right politics and its impact on public health [12], it is fundamental to analyze political discourses related to health, particularly when they affect vulnerable populations.

Considering all these factors, this study has two main objectives. Firstly, it aims to evaluate the level of attention given to HIV in current parliamentary initiatives, identify their primary thematic areas, and analyze their distribution according

to the ideologies of parliamentary groups. Secondly, it seeks to conduct a Critical Discourse Analysis (CDA) of political discourses related to HIV to understand the meanings and representations that are (re)produced in them. To achieve these objectives, this study investigated the case of the 12th Legislature of the Madrid Regional Parliament.

The hypothesis derived from the theoretical framework presented is that most of HIV parliamentary initiatives will be proposed by left-wing political parties, which will highlight social justice and equity in their discourse, challenging stigmatizing perceptions of HIV, while right-wing parties will focus on HIV as a threat to security and individual responsibility.

## Materials and methods

A mixed-methods quantitative and qualitative study was conducted to analyze parliamentary initiatives registered during the 12th Legislature of the Madrid Regional Parliament (2021–2023).

All parliamentary registered initiatives during the Legislature were systematically included in the analysis, as well as parliamentary records of the debates of those initiatives in parliamentary sessions. All the documents were publicly available at the parliament webpage.

Descriptive analyses were performed to quantify the total number of HIV-related initiatives and to determine their proportion relative to the overall legislative activity. Additionally, the temporal distribution of initiatives was examined. In initiatives debated in plenary sessions, attention was paid to the dates chosen for their discussion, particularly whether they coincided with June 28th (LGBTIQ+ Pride Day) or December 1st (World AIDS Day).

The quantitative analysis aimed to measure the extent and characteristics of parliamentary attention to HIV during the period studied. To this end, all parliamentary initiatives presented during the legislature were systematically reviewed using publicly available official records and session logs of the regional parliament. A database was constructed by identifying, compiling and coding those related to HIV. These initiatives were classified by the type of parliamentary initiative, thematic focus and political group affiliation.

Type of parliamentary initiative: distinguishing between 1) those debated orally in plenary sessions or committees: Non-Legislative Propositions (PNL, by its initials in Spanish), Appearances, Oral Questions for Response in Plenary Sessions (PCOP), Oral Questions for Response in Committee (PCOC), and 2) those exclusively processed in writing, Written Questions (PE), Citizens' Questions (PRECI), Requests for Information (PI), and initiatives initially meant to be oral that were not debated at the end of the legislature.

Thematic focus: each initiative was assigned one or more thematic axes based on its HIV-related content (pre-exposure prophylaxis, general prevention, anti-retroviral treatment and universal access, general policy and organization, stigma, contracts audits and request for incidence data). Two researchers independently assigned the thematic axes to the initiatives, remaining blind to the ones proposed by the other researcher. There were no disagreements between both researchers on the thematic axes.

Political group affiliation: The ideologies of the parties that presented the initiatives were also analyzed. The public political survey conducted by the Center for Sociological Research after the 2021 Madrid regional elections was used in order to establish the ideology of the various political parties within the left-right spectrum [28]. According to this survey, Madrid citizens perceived two left-wing political parties forming parliamentary groups in that Legislature, one center-left party, and two right-wing parties.

The purpose of this quantitative component was to assess the visibility and political prioritization of HIV in the parliamentary agenda, identify trends by ideologies of the parliamentary groups, and establish a context for the subsequent qualitative analysis.

CDA was chosen as the qualitative analysis method because it allows to focus on the relationship between language, power, and ideology; a relationship that is key in political and parliamentary discourse [29]. CDA allows for establishing the relationships between discourses and how the use of social power, dominance, and inequality are practiced, reproduced, and occasionally countered by texts and speech in social and political contexts [30,31]. Following this, parliamentary

discourses are not only a vehicle for communication but also tools to carry ideology (re)producing ideological positions and legitimizing power structures.

CDA is particularly suited for this study as it analyses how discourses can contribute to (re)produce or challenge social inequalities and power imbalance, showing how images around HIV intersect with political ideologies and policy decisions. This way, analysis can focus on whether parliamentary discourses contribute to or resist HIV-related stigma and inequality.

CDA has been applied to analyze parliamentary debates, institutional speeches and public policy discussions, as it combines text analysis with an understanding of the sociopolitical context where those texts are produced, undercovering ideological perspectives within the language [32–34].

CDA was performed on parliamentary records considering the ideology of the political parties and analyzing the political principles related to the themes and sub-themes developed. Two researches developed the themes and sub-themes and discussed them together in case of discrepancy. In the case of further discrepancy, the inclusion of a third researcher in the discussion was planned, but it was not needed as minor differences in categorization were discussed and resolved through consensus.

This study was reported according to the Standard for Reporting Qualitative Studies (SRQR) [35]. SRQR checklist can be found in Supplementary S1 Table.

### Ethics and consent

This study was conducted using publicly available data, which did not require specific ethical approval or informed consent, as the data are parliamentary records freely accessible to the public. Therefore, no additional ethical considerations were necessary for this research.

### Positionality and reflexivity statement

Authors, as public health professionals and researchers, consider that social determinants of health are a key component in the understanding of health, and strongly stand for equity and social justice as a fundamental part of public health. In addition, they support that any public health strategy must be grounded on human rights and ensure non-discrimination and protection of vulnerable populations.

## Results

### Number of initiatives and prioritized themes

During the 12th Legislature, 40,644 initiatives were presented in the Madrid Regional Parliament, of which 36 (0.09%) were related to HIV. Among these, 83.3% were merely written initiatives, while only 6 were debated orally in plenary sessions: 1 Non-Legislative Proposition that was processed and rejected, 4 Oral Questions for Response in Plenary Sessions, and 1 Appearance by the Health Counsellor. One PRECI was the only initiative presented by citizens instead of a parliamentary group, but it was not admitted by the Parliament's table as they considered that it did not observe "parliamentary courtesy" (Table 1).

Three PNLs were registered, but only one was debated in plenary sessions. The other two PNLs lapsed as the presenting parliamentary group did not request their debate. The PNL that was processed discussed LGBTIQ+ rights in general, with only one point related to HIV, and it was rejected in the vote.

The most prioritized themes among the presented initiatives were HIV pre-exposure prophylaxis (PrEP) and general prevention, with 15 and 10 initiatives respectively, comprising 66.7% of the total. Six initiatives discussed antiretroviral treatment (ART) and universal access to it, five addressed general HIV policy and organization, and only 4 (11.1%) were related to stigma. Additionally, 4 initiatives sought to audit public contracts with HIV-related organizations (Table 1 and Fig 1).

**Table 1. Madrid Regional Parliament 12th Legislature HIV-related initiatives according to type, theme, ideology of the parliamentary group that presented them and plenary session date.**

| | | Type of initiative | | | | | | |
| --- | --- | --- | --- | --- | --- | --- | --- | --- |
| | | Written initiatives | | | | Oral initiatives | | |
| | | PE | PI | PNL(l) | PRECI* | PCOP | PNL(r) | A |
| **TOTAL** | **36** | **14** | **13** | **2** | **1** | **4** | **1** | **1** |
| **Themes** | | | | | | | | |
| PrEP | **15** | 3 | 8 | 1 | 1 | 1 | 1 | |
| General prevention | **10** | 5 | | | | 3 | 1 | 1 |
| ART and universal access | **6** | 4 | | 1 | | 1 | | |
| General policy and organization | **5** | 3 | | 2 | | | | |
| Stigma | **4** | 2 | | 1 | | | | 1 |
| Contracts audits | **4** | | 4 | | | | | |
| Request for incidence data | **1** | | 1 | | | | | |
| **Parliamentary group ideology** | | | | | NA | | | |
| Left-wing | **19** | 12 | 3 | 1 | | 2 | 1 | |
| Center-left | **16** | 2 | 10 | 1 | | 2 | | 1 |
| Right-wing | **0** | | | | | | | |
| **Plenary session date** | | NA | NA | NA | NA | | | |
| Around December 1st | **4** | | | | | 3 | | 1 |
| Around June 28th | **1** | | | | | | 1 | |
| Other | **1** | | | | | 1 | | |

PE: Written Question. PI: Request for information. PNL(l): Non-Legislative Proposition (lapsed). PRECI: Citizens' Question. PCOP: Question for Response in Plenary Session. PLN(r): Non-Legislative Proposition (rejected). A: Appearances. PrEP: HIV pre-exposure prophylaxis. ART: anti-retroviral treatment. NA: not applicable. *The PRECI was not admitted.

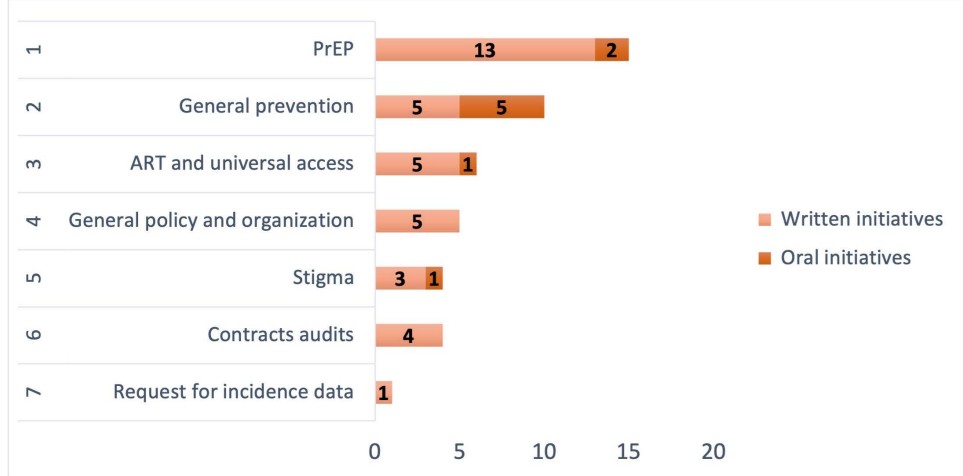

**Fig 1. Number of oral and written HIV-related parliamentary initiatives according to theme.** PrEP: pre-exposure prophylaxis. ART: anti-retroviral treatment.

Regarding the dates of the plenary sessions where the six oral initiatives were debated, four (66.7%) were scheduled around December 1st, World AIDS Day, and one around LGBTIQ+ Pride. Therefore, only one initiative was brought to the plenary session based on programmatic priorities rather than commemorative dates (Table 1).

All the initiatives were presented by left-wing or center-left parliamentary groups, with 19 and 16 initiatives respectively. No HIV-related initiatives were presented by right-wing parliamentary groups (Table 1).

**Critical discourse analysis by parliamentary groups ideology**

During the PNL debate on LGBTIQ+ rights, only two parties addressed HIV: one left-wing party advocated for early care, while one right-wing party opposed PrEP public funding. During the Health Counsellor's Appearance, both sides made statements against stigma, but on two occasions, the right-wing parties referred to HIV as a *disease* and lamented the *loss of fear* towards HIV.

From a broad view, the discourses mainly focused on prevention, generally approached from a perspective of *fear* of infection and *(in)security*, rather than on interventions related to treatment, combating stigma, or other issues affecting people living with HIV. Several parliamentary groups of different ideologies praised preventive measures, particularly the implementation of PrEP and efforts to improve its accessibility by reducing waiting lists and decentralizing its administration:

*"More and more tests are being conducted, and PrEP dispensation is increasing"*

*"[...] decentralizing the dispensation of PrEP is a measure recognized by all experts to combat and prevent HIV, but unfortunately is still not accessible to the people of Madrid."*

*"We continue to work on the dispensation of pre-exposure prophylaxis"*

In line with preventive measures, condom use was discussed as a specific means of protection against HIV and other sexually transmitted infections, framed from a perspective of *protection* against a *risk*:

*"So we insist: condoms are the best method for preventing HIV transmission"*

The political discourse also focused on early diagnosis strategies targeting populations perceived as high-risk or vulnerable groups, and the continuity of care:

*"The promotion of [...] early detection, both in hospital emergency rooms and in primary care settings, to identify hidden infections"*

*"[...] early diagnosis of HIV infection, [...] improving access for populations particularly vulnerable to HIV."*

*"[...] we will implement an HIV early detection program [...] to identify hidden or late-diagnosed infections in patients with a certain clinical profile who visit clinics for other reasons, for their study and clinical follow-up."*

Regarding the right-wing parliamentary groups, two important thematic axes were identified: *fear* and the perception of *risk*. In this case, awareness was associated with the idea of increasing the perception of fear towards HIV:

*"it is essential to focus on risk factors"*

The association of "HIV" with the term "disease" was common in the discourses of these parliamentary groups:

*"[…] preventing HIV infection in people who are at high risk of contracting the disease"*

Despite the apparent effort to maintain objectivity by citing data and official organizations, the right-wing parties' ideology is evident in their terminology. This includes pointing out *individual responsibility*, moral judgments on factors such as the number of sexual partners or the age at the first sexual relation, as well as portraying as negative the supposed loss of fear of HIV among the population:

*"[A report from the Ministry of Health] identifies the causes of the increase in sexually transmitted diseases as the greater number of sexual relations and partners, the loss of fear of HIV [...]"*

*"We need to warn about the early onset of sexual relations, about promiscuity."*

The HIV-related discourse of these right-wing parliamentary groups also projected a strong negative emotional burden. HIV was directly associated with traditionally oppressed sexual minorities, referred to as "privileged" by these parliamentary groups, and individuals were presented as rational agents who expose themselves to the risk of infection. This approach can activate stigmatization processes not only against people living with HIV but also against certain sexual practices and the LGBTIQ+ community. This part of their discourse is strongly linked to positions against PrEP — including a free, aware and autonomous sexuality — and, particularly, against its funding by the public healthcare system. The right-wing discourse directly appealed to the general population and, specifically, the working class, seeking to position them against sexual minorities:

*"the risk compensation theory is known to lead to greater relaxation [of preventive measures]"*

*"[…] They demand public funding instead of avoiding risky practices, which means that taxpayers are paying for the irresponsibility of privileged groups. The normal thing would be for each person to pay for their own party […]"*

*"What can the people of Madrid, who are standing in food lines or unable to pay their mortgages, feel knowing that their tax money is being spent on inseminating lesbians? Or how might they feel if they don't receive their benefits, knowing that their money is going to pay for the party of irresponsibility of groups that don't protect themselves in their risky practices?".*

In contrast, the left-wing and center-left parliamentary groups stood up for the use of PrEP and its public funding, emphasizing prevention while shifting the focus away from fear:

*"PrEP is a preventive treatment that is included in the public healthcare system and significantly reduces cases of HIV infection"*

On the other hand, a central theme in the discourse of the left-wing and center-left parliamentary groups was the emphasis on *discrimination and stigma as significant issues* and ensuring the inclusion of people living with HIV, prioritizing social justice. They echoed, for example, the UNAIDS 2030 targets:

*"Ladies and gentlemen, Mr. Counsellor: 95-95-95; zero discrimination"*

*"Undetectable = Untransmittable"*

*"[...] the greatest risk that this infection currently poses, Mr. Counsellor, is stigma"*

They also referenced the stigmatizing content of the right-wing speeches and a neo-Nazi demonstration that had recently taken place in Madrid's queer district with shouts against people living with HIV:

*"I wonder if some of these speeches made here [...] and all those hate speeches [...] might have something to do with the fact that few resources are allocated against stigma"*

*"[...] we have to end the virus, yes, but we also have to end any kind of stigma associated with HIV, and for that, we have to shut the door on those who walk the streets of Madrid shouting 'Out, AIDS carriers! '"*

Finally, as a matter of equity, left-wing parliamentary groups prioritized the need to provide universal access to appropriate treatments for people living with HIV, regardless of their administrative status, criticizing the regional government for excluding these individuals from the healthcare system:

*" You have turned Madrid into the only region, I repeat, the only region where people are excluded from antiretroviral treatment solely based on their administrative status."*

*"The region of Madrid is the only autonomous community in all of Spain where they are being denied medication due to their administrative status"*

*"But the most serious [...] is what I mentioned about people in irregular administrative status [...], who are being denied treatment"*

The themes and sub-themes developed in CDA are summarized in Table 2.

## Discussion

Firstly, it is notable that HIV is almost absent in the parliamentary activity of the Madrid Regional Parliament, and when it is present, it is mostly on specific dates like World AIDS Day (December 1st). Therefore, political activity around HIV in this parliament is driven more by commemorative dates than by prioritizing the needs of the community as they arise.

Secondly, it stands out that the vast majority of parliamentary initiatives focus on PrEP and other preventive methods, that is, on people who do not live with HIV. Initiatives related to issues that directly affect people living with HIV, such as ART or stigma, were relegated to the background. Consequently, the priority of the parliament, even when discussing HIV, is on people who do not live with HIV, despite the fact that it is estimated that between 136,436 and 162,307 people live with HIV in Spain [36]. This information is commonly disregarded, but the fact that most of the political actions and campaigns on HIV do not focus on people living with HIV exemplifies how vulnerable and stigmatized communities are rarely a political priority [37].

Preventive measures are of course fundamental, but focusing HIV-related discourses predominantly on prevention, especially when using hygienist frameworks, could reinforce stigmatizing processes towards people living with HIV, as it

**Table 2. Themes and sub-themes developed in the critical discourse analyses of parliamentary HIV-related discourses, related political principles and political parties' ideology.**

| Themes and sub-themes | Related political principles | Parties' ideology* |
|---|---|---|
| **Prevention** | Security and equity | Both |
| Improving PrEP implementation | Equity | Left-wing |
| Protection against risks | Security | Right-wing |
| Prioritization of vulnerable groups | Social justice | Left-wing |
| **Individual responsibility** | Individualism/ neoliberalism | Right-wing |
| Moral judgements | Individualism/ authoritarianism | Right-wing |
| Vulnerable groups signaling | Authoritarianism | Right-wing |
| **Discrimination and stigma prevention** | Social justice | Left-wing |
| **Equity in access to treatments** | Equity | Left-wing |

*\*Left-wing includes center-left. PrEP: HIV pre-exposure prophylaxis*

continues to be seen as a discrediting element that marks some and makes them different from the rest of the population [38]. As Villaamil mentions [39], HIV prevention measures targeting people not living with HIV, by emphasizing a limited view of the individual as a rational agent, may produce a moral economy of protection against others perceived as dangerous. Previous studies have problematized how institutional discourses and institutional public health campaigns use fear, blame attribution and frame HIV as a threat to security [20,27].

This study examines the political discourses about HIV and their potential stigmatizing effects on people living with HIV, utilizing the stigmatizing images that these discourses evoke from their cultural and ideological frameworks. Unlike previous studies, this research analyzed political discourses in parliament and parliamentary initiatives as a whole. Focusing on parliamentary contexts is particularly relevant since these representations significantly influence public opinion, to the extent that political hate speech has been linked to an increase in hate crimes [40]. Furthermore, these parliamentary discourses set the agenda and influence decisions on future social and health policies.

When analyzing the parliamentary initiatives related to HIV in the Madrid Regional Parliament, it stands out that all of them were presented by left-wing or center-left groups. In the CDA, two thematic axes can be distinguished within the right-wing groups' discourses: fear and the risk of infection, which activate discrediting elements regarding people living with HIV. Conversely, within the left-wing and center-left groups, the thematic axes of the discourse focus on addressing the stigma and discrimination faced by people living with HIV. Additionally, these latter parliamentary groups also emphasize the importance of universal access to ART.

These findings align with previous evidence that point out that left-wing parties tend to prioritize social determinants of health and focus on equity and social justice, for instance challenging stigma and discrimination and, thus, providing a public health perspective [11]. Findings also support evidence that right-wing parties prioritize neoliberalism, individualism and the concept of security when discussing public health issues [10,12]. The use of fear and the sense of insecurity are strategies commonly used in right-wing discourses in order to maintain stigma and privileges, as well as to normalize exclusion [13]. Blame attribution and focus on individual responsibility, part of right-wing discourses, are also direct consequences of their neoliberalist ideology, which revolves around individualism as opposed to social determinants of health [41]. This view intentionally omits the fact that most health inequalities are caused by social determinants and not by individual behaviors [37,42].

It is important to understand how certain statements or information produce representations about HIV. Both the literal content of these discourses and the latent meanings activated by political parties, along with their implications, must be considered in the analysis. According to Goffman's definition [38], stigma is not only related to "discredited" people (in this case, people living with HIV) but also to those "discreditable" (for example, the LGBTIQ+ community, seen as carriers of "risky" sexual practices). In the analyzed discourses of right-wing parliamentary groups, it can be observed how the sexuality of certain groups is marked as discreditable as potentially "contagious", thereby extending stigmatization to them. Part of these discourses sought to position the population against the LGBTIQ+ community, labeling them as "privileged" and ignoring the historical and current discrimination they face. This right-wing signaling of vulnerable populations is in line with described far-right authoritarianism [12].

This ideological division around HIV is not new, although to our knowledge it has not been studied in parliamentary contexts before. Traditionally, the greatest discrimination against people living with HIV has come more from the political right-wing. From the infamous statements of Republican politicians like Louie Welch and his "if you want to end AIDS, shoot the queers" [43], to the Francoist policies that persecuted LGBTIQ+ people in Spain, and whose ideological frameworks have been inherited by contemporary right-wing political parties. Previous studies have described that political ideology is related to HIV stigma at an individual level, with right-wing or conservative individuals being more likely to stigmatize [44,45].

Future studies should analyze how the perspective of risk and the exaggerated perception of fear in other national and international parliamentary contexts can activate processes of discreditation and stigmatization of people living with HIV, as well as increased surveillance of certain groups or sexual practices.

Sontag pointed out how certain diseases or infections contain metaphorical traps with negative social consequences for those living with them, making it necessary to dismantle these metaphors [26]. The strategy of deconstructing certain imaginaries around HIV allows us to move away from ideas like fear, contagion, punishment, or shame, which only complicate the lives of people living with HIV. In this sense, parliamentary discourses have great power to end HIV stigma and its generation of a "spoiled identity".

## Strengths and limitations

This study offers a relevant contribution to understanding how HIV-related issues are addressed in parliamentary discourse, with a mixed-methods approach. Its main strength lies in the comprehensive and systematic analysis of all initiatives presented during an entire legislative term, allowing us to identify how frequently HIV is mentioned, in what context, and by which political parties.

One limitation of the study is the relatively low number of HIV-related initiatives identified (36 out of 40,644), which reflects the limited political attention given to the issue and reduces the possibility of more complex quantitative analysis. Another limitation is that the study focuses only on the Madrid Regional Parliament, which may affect the generalizability of the results to other territories or institutional settings.

However, Madrid is not just any region. It is the political, administrative, and media capital of Spain, and often plays a central role in shaping national political agenda and debates. Therefore, the trends identified here may reflect broader dynamics present in other regions or at the national level. In addition, the dynamics described in relation to the political parties' ideologies align with international evidence on public health discourses in other countries. These discourses are not exclusive to Madrid, but form part of a wider international trend that poses challenges to equity-based and rights-oriented public health policies. In this sense, the study contributes to a broader reflection on how political ideologies influence public health beyond the regional context.

CDA, as any methodology that requires interpretation and social context, has the limitation of having some risk of subjectivity. However, the use of official parliamentary transcripts as the source of data ensured reproducibility. On the other hand, CDA has the strength of focusing on the sociopolitical dimensions of discourse, which perfectly suits the analysis of parliamentary discourses, where CDA has a robust track record.

## Conclusions

Issues around HIV are hardly ever discussed in the Madrid Regional Parliament, and mostly focus on prevention and people not living with HIV. When HIV is discussed, the ideology of the parliament political parties has a crucial role on discourses. The discourse analysis showed that right-wing parliamentary groups mostly frame HIV in terms of fear, risk of infection and (in) security. Their discourse included stigmatizing language, pointed individual responsibility in line with neoliberalist ideology, and comprised statements that emphasize negative aspects and sought to raise social alarm about infection. Right-wing parties also associated risk and blame with sexual minorities and opposed public funding for PrEP. Conversely, the discourses of left-wing or center-left parliamentary groups recognized stigma as a problem and emphasized the importance of universal access to ART, which aligns with their prioritization of social justice and equity from a rights-based public health perspective.

These findings contribute to the literature pointing out the intersection of political ideology and discourse with public health, analyzing the manifestation of ideology in discourses around a vulnerable and stigmatized population, such as people living with HIV. Focusing on a parliamentary setting fills a previous gap on the HIV stigma research.

Future research should critically analyze policy initiatives and parliamentary discourses around HIV in other national and international legislative bodies to explore the generalizability of the patterns found.

This has important practical implications as parliamentary debates not only reflect but can also shape public opinions and legitimize discrimination. Certain political representations of HIV can trigger stigmatizing processes, associating people living with HIV with risk and contagion, and lead to blame and exclusion.

The findings support that not all ideologies are equally beneficial to achieve public health objectives of equity and justice in health, which are wider represented in left-wing discourses. Therefore, parliamentary actors and public health policymakers, especially right-wing parties, must become aware of the material consequences and stigmatizing risks of their ideology-based discourses, and focus on implementing public policies aimed at combating stigma and discrimination from a human rights-based public health approach.

## Supporting information

**S1 Table. SRQR checklist.** This checklist contains the Standard for Reporting Qualitative Research (SRQR) items. (DOCX)

## Acknowledgments

This is an independent research. The views presented in this article are those of the authors and do not necessarily reflect positions of their affiliated institutions.

## Author contributions

**Conceptualization:** Paule Gonzalez-Recio, Édel Granda.

**Data curation:** Paule Gonzalez-Recio, Édel Granda.

**Formal analysis:** Paule Gonzalez-Recio, Édel Granda.

**Investigation:** Paule Gonzalez-Recio.

**Methodology:** Paule Gonzalez-Recio, Édel Granda.

**Project administration:** María José Belza.

**Supervision:** Paule Gonzalez-Recio, Raquel Barba-Sánchez, Juan-Miguel Guerras, María José Belza.

**Validation:** Paule Gonzalez-Recio, Raquel Barba-Sánchez, María José Belza.

**Visualization:** María José Belza.

**Writing – original draft:** Paule Gonzalez-Recio, Raquel Barba-Sánchez.

**Writing – review & editing:** Paule Gonzalez-Recio, Raquel Barba-Sánchez, Juan-Miguel Guerras, Sara Moreno-García, Alex Iglesias, María José Fuster-RuizdeApodaca, Esther García Expósito, David Palma, María José Belza.

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
