## [Decision Letter · Decision Letter 0]

9 Jul 2025

Dear Dr. Barba Sánchez,

We look forward to receiving your revised manuscript.

Kind regards,

Sylvester Chidi Chima, M.D., L.L.M, LLD

Academic Editor

PLOS ONE

Reviewers' comments:

Reviewer's Responses to Questions

**Comments to the Author**

1. Is the manuscript technically sound, and do the data support the conclusions?

Reviewer #1: Yes

Reviewer #2: Partly

2. Has the statistical analysis been performed appropriately and rigorously?

Reviewer #1: Yes

Reviewer #2: N/A

3. Have the authors made all data underlying the findings in their manuscript fully available?

Reviewer #1: Yes

Reviewer #2: Yes

4. Is the manuscript presented in an intelligible fashion and written in standard English?

Reviewer #1: Yes

Reviewer #2: Yes

Reviewer #1: This is a well-written manuscript on an important aspect. I have a few comments.

Authors need to describe the analysis process in more detail. For example, the manuscript currently states that ‘The lexical, grammatical, and textual dimensions of the discourse formulation of each parliamentary group were considered’. What did this process involve? Author contributions state that PGR conducted the analysis and interpretation of the data together with EG. What did data analysis and interpretation involve? Were there any discrepancies or disagreements? If so, how were these resolved? Methods also state –‘Additionally, elements impacting processes of stigmatization of people living with HIV and sexual minorities were considered.’ What did this involve? How did authors arrive at the themes for example?

Discussion needs to include a strengths and limitations section/paragraph. For example, one limitation could be that the results are informed by a relatively small number of initiatives 36/40,644. In addition authors could include a positionality/reflexivity statement.

Minor issues

initiatives in in this context (delete 2nd “in”)

seeking to position them against…[them]?

Methods heading in body of paper should be ‘Materials and Methods

Mixed-method(s)?

Just wondering if there is any way of referencing the quotes.

Include a space before references – e.g., us [1]

Reviewer #2: I have carefully reviewed the manuscript entitled “From Fear of Infection to Awareness Against Stigma: A Mixed-Methods Analysis of Discourses on HIV in Parliamentary Contexts.” The study addresses a timely and significant topic, particularly in light of the ongoing stigmatization of HIV and the relatively limited research on the role of legislative discourse in shaping public health narratives. While the manuscript demonstrates originality and relevance, there are several aspects that require further development and clarification. Therefore, I recommend that the authors undertake substantial revisions to enhance the overall quality and rigor of the work prior to its consideration for publication. My comments are as follows:

1) The manuscript would benefit from a dedicated literature review section, distinct from the introduction, to provide a more comprehensive overview of recent and relevant scholarly work. A separate literature review should incorporate the most up-to-date research related to the intersection of political ideology and public health, particularly in the context of HIV.

Moreover, the authors should offer a more explicit discussion of their study's contribution. While they state that, to their knowledge, no prior research has examined the relationship between parliamentary discourses on HIV and political ideologies, this claim requires substantiation—such as evidence of a systematic literature search. The identification and articulation of the research gap should be strengthened through engagement with similar studies, not only in HIV but also in broader analyses of how political ideologies influence health outcomes or other issues. Referencing relevant comparative studies from other national or regional contexts would further reinforce the manuscript’s originality and significance.

2) The manuscript would benefit from a more explicit articulation of the theoretical framework underpinning the study. The authors should clearly outline how existing theories explain the influence of political ideology on public discourse, particularly in relation to health issues such as HIV.

Drawing on relevant literature, the manuscript should specify the theoretical assumptions guiding the analysis and clearly state the main research question or hypotheses derived from this foundation. This would help to clarify the conceptual underpinnings of the study and establish a more coherent analytical framework.

In addition to social representation theory, partisan theory should also be examined by the authors.

Whether incorporated within the introduction or presented as a distinct section, a well-defined theoretical framework would not only enhance the study’s rigor but also position it more firmly within the broader academic discourse on political ideology and public health.

3) The study claims to employ both qualitative and quantitative analyses; however, it presents only qualitative analysis in the form of discourse analysis, with no clear evidence of quantitative analysis. The quantitative component should be explicitly defined, and the methodology and results clearly presented to ensure methodological coherence and transparency.

4) The Methods section would benefit from greater detail, particularly regarding the rationale for selecting Critical Discourse Analysis (CDA) as the primary methodological approach. The authors should clearly justify the choice of CDA over alternative methods, explaining how it aligns with the research objectives and is well-suited to analyzing the specific characteristics of parliamentary discourse. Additionally, the reliability and applicability of CDA in this context should be supported with appropriate references from the literature. A more robust explanation of the research design, including the methodological strengths and potential limitations of CDA, would enhance the study's transparency and rigor.

5) In the Discussion section, the authors are encouraged to compare and contrast their findings with those of previous studies in the field. This would help to contextualize the results, underscore the originality of the research, and clarify its contribution to the existing body of knowledge.

6) The conclusion section should be expanded to offer a more comprehensive analysis of the implications of the study’s findings. The authors should clearly articulate the study’s contributions to the existing literature and highlight its practical significance, particularly in relation to policymaking and public health discourse. Additionally, a critical reflection on the study’s limitations is recommended, along with suggestions for future research directions that could further explore the relationship between political ideology and health-related discourse.

7) The generalizability of the findings is limited to one regional context. Although this is acknowledged, a more explicit discussion of how these insights may or may not extend to national or other international legislative bodies would strengthen the paper.

8) The manuscript contains occasional grammatical errors and typographical mistakes, such as the repetition of 'in' in the phrase "in in this context" found in the abstract.

**Do you want your identity to be public for this peer review?** For information about this choice, including consent withdrawal, please see our Privacy Policy

Reviewer #1: No

Reviewer #2: **Yes: ** Prof. Dr. Rasim Yilmaz

---

## [Author Response · Author response to Decision Letter 1]

13 Aug 2025

Response to Reviewers and Editor

First of all, we would like to thank both peer reviewers and PlosOne editorial team, as we believe the paper has improved in clarity and quality after this review process.

1. Please submit a completed SRQR (Standard for Reporting Qualitative Studies) Checklist or in the alternative a COREQ checklist which clearly shows the key elements of qualitative analysis in this study, including which researchers conducted the primary and secondary analyses of the qualitative data, as well as major 'themes' and 'sub-themes' derived from the data analysis.

SRQR checklist has been submitted as Supplementary Table S1. The following text has been added to the methods section:

This study was reported according to the Standard for Reporting Qualitative Studies (SRQR) (CITA). SRQR checklist can be fount in Supplementary Table S1.

Consider summarizing the 'themes and sub-themes' derived from the qualitative analysis into a separate table in the results section.

This table has been added at the end of the results section. We would like to thank this suggestion as we believe it improves the clarity of the analysis.

Table 2. Themes and sub-themes developed in the critical discourse analyses of parliamentary HIV-related discourses, related political principles and political parties’ ideology.

Themes and sub-themes Related political principles Parties’ ideology*

Prevention Security and equity Both

Improving PreP implementation Equity Left-wing

Protection against risks Security Right-wing

Prioritization of vulnerable groups Social justice Left-wing

Individual responsibility Individualism / neoliberalism Right-wing

Moral judgements Individualism / authoritarianism Right-wing

Vulnerable groups signaling Authoritarianism Right-wing

Discrimination and stigma as issues Social justice Left-wing

Equity in access to treatments Equity Left-wing

*Left-wing includes center-left. PrEP: HIV pre-exposure prophylaxis

2. Kindly expand on the 'theoretical framework' underpinning this study, and justify the reasoning behind the choice of the "Critical Discourse Analysis (CDA) as the primary methodological approach" for this study as suggested by Reviewer 2.

Theoritical framenwork has been expanded and articulated in detail in the introduction, with a deeper explanation of how political ideologies shape political discourses on health related issues, including partisan theory as suggested by Reviewer 2. These paragraphs have been added/modified in the introduction:

Political ideology, can then be understood not merely as a preference on certain policies, but as a systemic symbolic tool that shapes social meanings using policy and discourse [4,5]. Political ideologies deeply influence how social issues are framed and how stigma is (re)produced or challenged and can have a profound impact on health.

Political parties are also a fundamental part of modern democracies and the production of political discourses and identity politics. According to partisan theory, the ideological orientation of political parties shapes their positions, priorities and communicative strategies, while partisan attachments have the ability to affect reasoning, perceptions and policy evaluation by individuals [6,7]. Parties ideology not only determines the priorities of politics, but also how policy is brought to the political arena, determining, for example, if health is framed as a matter of human rights, the result of individual choices and behavior, or as a threat to public safety [8,9].

Left-wing parties are more likely to prioritize key public health issues, such as equity, social justice and social determinants of health, while right-wing parties are more like to understand health from a neoliberalist perspective emphasizing individual responsibility, national security and biomedical aspects of health [10,11]. Growing far-right policies have been pointed out because of how their neoliberalist, authoritarian, anti-science and anti-equality ideology impact public health maintaining privileges of those in power and worsening inequalities [12,13]. Thus, political ideology and discourse not only represent ideas and thoughts, but have real consequences on the lives and health, particularly, of vulnerable populations.

In parliamentary contexts, the main arena of political parties, ideology driven discourses are particularly relevant because of their potential legal and normative repercussions, and their dual role in reflecting and influencing public discourse and constructing collective representations and understandings of health issues [13], such as HIV.

The methods section regarding CDA has been extended in order to add a more robust explanation on the selection of these methodology and why it is suited for this research, including exmples of previous uses of CDA by other researchers:

CDA was chosen as the qualitative analysis method because it allows to focus on the relationship between language, power, and ideology; a relationship that is key in political and parliamentary discourse [29]. CDA allows for establishing the relationships between discourses and how the use of social power, dominance, and inequality are practiced, reproduced, and occasionally countered by texts and speech in social and political contexts [30,31]. Following this, parliamentary discourses are not only a vehicle for communication but also tools to carry ideology (re)producing ideological positions and legitimizing power structures.

CDA is particularly suited for this study as it analyses how discourses can contribute to (re)produce or challenge social inequalities and power imbalance, showing how images around HIV intersect with political ideologies and policy decisions. This way, analysis can focus on whether parliamentary discourses contribute to or resist HIV-related stigma and inequality.

CDA has been applied to analyze parliamentary debates, institutional speeches and public policy discussions, as it combines text analysis with an understanding of the sociopolitical context where those texts are produced, undercovering ideological perspectives within the language [32–34].

CDA was performed on parliamentary records considering the ideology of the political parties and analyzing the political principles related to the themes and sub-themes developed. Two researches developed the themes and sub-themes and discussed them together in case of discrepancy. In the case of further discrepancy, the inclusion of a third researcher in the discussion was planned, but it was not needed as minor differences in categorization were discussed and resolved through consensus.

This paragrapgh has been added to the strengths and limitations section:

CDA, as any methodology that requires interpretation and social context, has the limitation of having some risk of subjectivity. However, the use of official parliamentary transcripts as the source of data ensured reproducibility. On the other hand, CDA has the strength of focusing on the sociopolitical dimensions of discourse, which perfectly suits the analysis of parliamentary discourses, where CDA has a robust track record.

3. Please consider expanding the introduction section with a more detailed review of relevant literature 'to provide a more comprehensive overview of recent and relevant scholarly work."

We have included more references to previous works in the introduction, particularly underpinning the intersection of ideology and public health, as well as the scare researches around recent HIV political discourses:

Left-wing parties are more likely to prioritize key public health issues, such as equity, social justice and social determinants of health, while right-wing parties are more like to understand health from a neoliberalist perspective emphasizing individual responsibility, national security and biomedical aspects of health [10,11]. Growing far-right policies have been pointed out because of how their neoliberalist, authoritarian, anti-science and anti-equality ideology impact public health maintaining privileges of those in power and worsening inequalities [12,13]. Thus, political ideology and discourse not only represent ideas and thoughts, but have real consequences on the lives and health, particularly, of vulnerable populations.

[…]

As Sontag points out, the political and discriminatory use of metaphors about HIV, as well as its portrayal as a political symbol or allegory, has devastating consequences for those living with HIV [26]. Most of the previous research on institutional discourses around HIV has focused on public health governance and campaigns. Institutional discourses initially focused on the control of vulnerable populations and blame attribution, adding weight to HIV stigma [20]. According to previous research, more recent institutional public health campaigns have used fear and framed HIV as a threat to security [27] in a similar way of how right-wing politicians prioritize security and individual responsibility while ignoring social determinants of health [10,11,13].

4. Please clarify the quantitative and qualitative methods used in this study. "The quantitative component should be explicitly defined, and the methodology and results clearly presented to ensure methodological coherence and transparency." as suggested by Reviewer 2.

We have revised and expanded the Methods section to ensure greater methodological transparency and coherence. We have included the following new paragraphs to clearly outline the quantitative procedures used:

All parliamentary registered initiatives during the Legislature were systematically included in the analysis, as well as parliamentary records of the debates of those initiatives in parliamentary sessions. All the documents were publicly available at the parliament webpage.

Descriptive analyses were performed to quantify the total number of HIV-related initiatives and to determine their proportion relative to the overall legislative activity. Additionally, the temporal distribution of initiatives was examined. In initiatives debated in plenary sessions, attention was paid to the dates chosen for their discussion, particularly whether they coincided with June 28th (LGBTIQ+ Pride Day) or December 1st (World AIDS Day).

The quantitative analysis aimed to measure the extent and characteristics of parliamentary attention to HIV during the period studied. To this end, all parliamentary initiatives presented during the legislature were systematically reviewed using publicly available official records and session logs of the regional parliament. A database was constructed by identifying, compiling and coding those related to HIV. These initiatives were classified by the type of parliamentary initiative, thematic focus and political group affiliation.

Type of parliamentary initiative: distinguishing between 1) those debated orally in plenary sessions or committees: Non-Legislative Propositions (PNL, by its initials in Spanish), Appearances, Oral Questions for Response in Plenary Sessions (PCOP), Oral Questions for Response in Committee (PCOC), and 2) those exclusively processed in writing, Written Questions (PE), Citizens' Questions (PRECI), Requests for Information (PI), and initiatives initially meant to be oral that were not debated at the end of the legislature.

Thematic focus: each initiative was assigned one or more thematic axes based on its HIV-related content (pre-exposure prophylaxis, general prevention, anti-retroviral treatment and universal access, general policy and organization, stigma, contracts audits and request for incidence data). Two researchers independently assigned the thematic axes to the initiatives, remaining blind to the ones proposed by the other researcher. There were no disagreements between both researchers on the thematic axes.

Political group affiliation: The ideologies of the parties that presented the initiatives were also analyzed. The public political survey conducted by the Center for Sociological Research after the 2021 Madrid regional elections was used in order to establish the ideology of the various political parties within the left-right spectrum[28]. According to this survey, Madrid citizens perceived two left-wing political parties forming parliamentary groups in that Legislature, one center-left party, and two right-wing parties.

The purpose of this quantitative component was to assess the visibility and political prioritization of HIV in the parliamentary agenda, identify trends by ideologies of the parliamentary groups, and establish a context for the subsequent qualitative analysis.

These additions clarify the scope, procedure, and purpose of the quantitative analysis.

5. Please include a clear "statement of strengths and limitations" at the end of the Discussion section to address potential limitations such as "the results are informed by a relatively small number of initiatives 36/40,644", as well as limitations regarding generalizability of the findings from this study to other jurisdictions.

We have added a new section that addresses both the methodological limitations of the study and the broader context of its relevance.

Specifically, we now acknowledge the relatively small number of HIV-related initiatives identified and the limitations for the scope of the quantitative analysis. We also address the issue of generalizability, noting that although the study is focused on the Madrid Regional Parliament, this institution holds particular relevance due to Madrid is the political and media capital of Spain, often acting as a national agenda-setter.

We emphasize that the ideological discourses observed—particularly those of right-wing and far-right parties—are consistent with broader political trends identified across Europe and internationally. To support this point, we have incorporated references to recent public health literature (March et al., 2025) that describe how the rise of far-right ideologies is influencing health policy and undermining equity and inclusion. We believe this contextualization strengthens the broader relevance of the study and addresses the concern raised.

This is the new section of "Strengths and Limitations":

This study offers a relevant contribution to understanding how HIV-related issues are addressed in parliamentary discourse, with a mixed-methods approach. Its main strength lies in the comprehensive and systematic analysis of all initiatives presented during an entire legislative term, allowing us to identify how frequently HIV is mentioned, in what context, and by which political parties.

One limitation of the study is the relatively low number of HIV-related initiatives identified (36 out of 40,644), which reflects the limited political attention given to the issue and reduces the possibility of more complex quantitative analysis. Another limitation is that the study focuses only on the Madrid Regional Parliament, which may affect the generalizability of the results to other territories or institutional settings.

However, Madrid is not just any region. It is the political, administrative, and media capital of Spain, and often plays a central role in shaping national political agenda and debates. Therefore, the trends identified here may reflect broader dynamics present in other regions or at the national level. In addition, the dynamics described in relation to the political parties’ ideologies align with international evidence on public health discourses in other countries. These discourses are not exclusive to Madrid, but form part of a wider international trend that poses challenges to equity-based and rights-oriented public health policies. In this sense, the study contributes to a broader reflection on how political ideologies influence public health beyond the regional context.

CDA, as any methodology that requires interpretation and social context, has the limitation of having some risk of subjectivity. However, the use of official parliamentary transcripts as the source of data ensured reproducibility. On the other hand, CDA has the strength of focusing on the sociopolitical dimensions of discourse, which perfectly suits the analysis of parliamentary dis

---

## [Decision Letter · Decision Letter 1]

15 Sep 2025

From fear of infection to awareness against stigma: a mixed-methods analysis of discourses on HIV in a parliamentary context

PONE-D-25-26592R1

Dear Dr. Sánchez,

We’re pleased to inform you that your manuscript has been judged scientifically suitable for publication and will be formally accepted for publication once it meets all outstanding technical requirements.

Kind regards,

Sylvester Chidi Chima, M.D., L.L.M, LD

Academic Editor

PLOS ONE

Reviewers' comments:

Reviewer's Responses to Questions

**Comments to the Author**

Reviewer #1: All comments have been addressed

Reviewer #2: All comments have been addressed

2. Is the manuscript technically sound, and do the data support the conclusions?

Reviewer #1: Yes

Reviewer #2: Yes

3. Has the statistical analysis been performed appropriately and rigorously?

Reviewer #1: Yes

Reviewer #2: Yes

4. Have the authors made all data underlying the findings in their manuscript fully available?

Reviewer #1: Yes

Reviewer #2: Yes

5. Is the manuscript presented in an intelligible fashion and written in standard English?

Reviewer #1: Yes

Reviewer #2: Yes

Reviewer #1: All comments have been adequately addressed. Authors have paid attention to detail in the revised manuscript.

Reviewer #2: All comments have been addressed. All required questions have been answered and that all responses meet formatting specifications.

**Do you want your identity to be public for this peer review?** For information about this choice, including consent withdrawal, please see our Privacy Policy

Reviewer #1: No

Reviewer #2: **Yes: ** Prof. Dr. Rasim Yilmaz

---

## [Editor Report · Acceptance letter]

PONE-D-25-26592R1

PLOS ONE

Dear Dr. Barba-Sánchez,

I'm pleased to inform you that your manuscript has been deemed suitable for publication in PLOS ONE. Congratulations! Your manuscript is now being handed over to our production team.

Kind regards,

on behalf of

Professor Sylvester Chidi Chima

Academic Editor

PLOS ONE